# Size-Dependent Effects of Polystyrene Nanoparticles (PS-NPs) on Behaviors and Endogenous Neurochemicals in Zebrafish Larvae

**DOI:** 10.3390/ijms231810682

**Published:** 2022-09-14

**Authors:** Kyu-Seok Hwang, Yuji Son, Seong Soon Kim, Dae-Seop Shin, So Hee Lim, Jung Yoon Yang, Ha Neul Jeong, Byung Hoi Lee, Myung Ae Bae

**Affiliations:** Therapeutics & Biotechnology Division, Korea Research Institute of Chemical Technology, Daejeon 34114, Korea

**Keywords:** zebrafish, polystyrene, nanoparticle, behavior, electroencephalography, neurochemical, dopamine, BBB chip, microfluidic system

## Abstract

Microplastics, small pieces of plastic derived from polystyrene, have recently become an ecological hazard due to their toxicity and widespread occurrence in aquatic ecosystems. In this study, we exposed zebrafish larvae to two types of fluorescent polystyrene nanoparticles (PS-NPs) to identify their size-dependent effects. PS-NPs of 50 nm, unlike 100 nm PS-NPs, were found to circulate in the blood vessels and accumulate in the brains of zebrafish larvae. Behavioral and electroencephalogram (EEG) analysis showed that 50 nm PS-NPs induce abnormal behavioral patterns and changes in EEG power spectral densities in zebrafish larvae. In addition, the quantification of endogenous neurochemicals in zebrafish larvae showed that 50 nm PS-NPs disturb dopaminergic metabolites, whereas 100 nm PS-NPs do not. Finally, we assessed the effect of PS-NPs on the permeability of the blood–brain barrier (BBB) using a microfluidic system. The results revealed that 50 nm PS-NPs have high BBB penetration compared with 100 nm PS-NPs. Taken together, we concluded that small nanoparticles disturb the nervous system, especially dopaminergic metabolites.

## 1. Introduction

Plastics are synthetic organic polymers that can be molded by deformation with heat, and the production of plastics is constantly increasing around the world. However, plastics cause ecological problems, as they do not completely degrade and remain in the ecosystem for hundreds of years. The degradation of plastic debris into microparticles and nanoparticles has recently become a significant concern in aquatic environments [1,2]. They can be distinguished based on diameter, with particles less than 5 μm defined as microparticles, while biological, physical, and chemical factors degrade microparticles further into nanoparticles, with diameters of less than 100 nm [3].

Living organisms, including humans, can absorb nanoparticles through the air, water, skin, and food chain [4,5]. Furthermore, nanoparticles inhaled with air accumulate in the respiratory tract [2]. In general, larger microparticles are likely excreted through mucociliary clearance in the lung and feces in the intestine [6]. Their accumulation in and excretion from living organisms can be different depending on size, shape, dose, surface functionalization, and charge [6]. In particular, toxicity due to the accumulation of nanoparticles tends to increase with a decrease in nanoparticle size [7].

Since nanoparticles have been detected in living organisms, many studies have been published on their toxicity. Nanoparticles were found to increase reactive oxygen species (ROS) generation in epithelial HeLa and cerebral T98G cell lines [8]. In addition, 1321N1, HepG2, and HEK 293 incubation with nanoparticles induced cytotoxic effects causing damage to cell membranes [9]. Another study assessed that nanoparticles affected the viability and morphology of human gastric adenocarcinoma cells, leading to inflammation [10]. Unlike in vitro testing, the effects of nanoparticles on animals can be assessed in more diverse fields. Exposing *Caenorhabditis elegans* to nanoparticles led to an increase in locomotor behavior and damage to GABAergic and cholinergic neurons [11]. Exposing Sprague–Dawley rats to nanoparticles resulted in a significant migration of neutrophils into the lungs, pneumonia, and an increase in IL-8 expression in epithelial cells [12].

In this study, we employed a zebrafish (*Danio rerio*) model to determine the effects of polystyrene nanoparticles (PS-NPs) on the central nervous system. Zebrafish have been used for many years to study the toxicological effects of chemicals and drugs. Although they enable in vivo studies on a large scale, questions remain about how relevant these studies are to humans. However, approximately 70% of coding genes are conserved between humans and zebrafish [13]. Physiology such as cardiac electrophysiology and metabolic components are also well conserved [14]. The development of embryos is rapid, with organogenesis completed 72 h post-fertilization (hpf). After hatching, larvae can be used to assess locomotor activity due to spontaneous swimming displays [15]. Experimental methods using zebrafish larvae to determine neurotoxicity have been well described. Light/dark transitions and color preference testing have been used to evaluate the anxiety response from stimuli and decision making, respectively [16,17]. Moreover, the quantification of endogenous neurochemicals and electroencephalogram (EEG) analysis have been well established in zebrafish larvae to determine the correlation with behavioral patterns [18,19].

Since the harm caused by nanoparticles has been reported, various data determining the toxic effects of nanoparticles on zebrafish as a model organism have been published. However, the size of the nanoparticles, along with the exposure stage, duration, and concentration have varied [20]. The present study aimed to assess the potential toxicity of polystyrene nanoparticles (PS-NPs) in zebrafish larvae, focusing on the central nervous system. To this end, we exposed zebrafish larvae to two sizes of fluorescent PS-NPs, 50 and 100 nm, and analyzed behavioral tests and EEGs. Moreover, we quantified endogenous neurochemicals using liquid chromatography–mass spectrometry (LC-MS/MS) to determine the correlation with behavioral patterns. Finally, we compared the permeability of the blood–brain barrier (BBB) between the two PS-NPs using a BBB chip.

## 2. Results

### 2.1. Accumulation of Polystyrene Nanoparticles (PS-NPs) in Zebrafish Larvae

After the embryonic period (~72 hpf), zebrafish larvae initiate swim bladder inflation and free swimming [15]. As the yolk becomes smaller, feeding usually begins at 5 days post-fertilization (dpf) [21]. In this study, exposure to PS-NPs occurred at 5 dpf when the larvae began spontaneous food intake. Two types of PS-NPs, depending on diameter size, were exposed at different concentrations (0, 500, 750, 1000, 1250, 1500 ppm) for 72 h (Figure 1A). PS-NPs of 100 nm did not induce larval mortality in any concentrations. However, exposure to 50 nm PS-NPs resulted in the survival rate decreasing to 83% at 1250 ppm and 30% at 1500 ppm (Figure 1B). Therefore, it was determined that the effective concentration of both PS-NPs is 1000 ppm for further experiments.

In order to visualize the accumulation of PS-NPs in zebrafish larvae, we exposed *Tg(kdrl:mCherry)* zebrafish with *nacre* background to PS-NPs. This transgenic line expresses the mCherry protein under the control of the *kdrl* promoter in the endothelial lineage, resulting in a visualization of the vasculature system in living larvae [22,23]. As the *nacre* zebrafish is a mutation of *melanocyte-inducing transcription factor a* (*mitfa*) that regulates melanogenesis [24], we used it as a background to clearly visualize the brain region. PS-NPs are also fluorescently monodispersed with carboxylate that is detected at 441 nm (maximum excitation) and 486 nm (maximum emission). Unlike 100 nm PS-NPs, 50 nm PS-NPs were distributed into the intestine and blood vessels of zebrafish larvae (Figure 1D). The 50 nm PS-NPs were highly accumulated in the brain region especially, including the forebrain, optic tectum, cerebellum, and hindbrain (Figure 1C). These results suggested that PS-NPs size-dependently accumulate in the brains of zebrafish larvae through the intestine and blood vessels, affecting their survival rate.

### 2.2. Effects of Polystyrene Nanoparticles (PS-NPs) on Central Nervous System

In order to determine whether PS-NPs influence the nervous system, since 50 nm PS-NPs accumulate in the brain, we conducted two behavioral tests, light and dark transition and color preference, in the zebrafish larvae. Locomotor activity was assessed as the distance moved (mm) following light and dark stimuli. The 50 nm PS-NPs caused an increased locomotion following light and dark stimuli, whereas the 100 nm PS-NPs did not affect locomotor activity in any stimuli (Figure 2A).

The cognitive ability of zebrafish larvae can be evaluated through color preference testing due to an innate color discrimination capability [17]. We performed color preference tests using a two-channel (blue and yellow) chamber in this study. The untreated control group tended to prefer blue (74.2%) more than yellow (25.8%), whereas the 50 nm PS-NP group showed almost the same preference for blue (52.6%) and yellow (47.4%). The 100 nm PS-NP group (blue, 66.7%; yellow, 33.3%) also showed a similar preference pattern to the control group (Figure 2B).

Next, we profiled EEG spectrums in PS-NP-exposed zebrafish larvae using relative power spectral density analysis: slow oscillation (0.5–1 Hz), pure delta (1–4 Hz), delta (0.5–4 Hz), theta (4–8 Hz), alpha (8–13 Hz), beta 1 (13–20 Hz), beta 2 (20–30 Hz), beta (12–30 Hz), and slow gamma (30–55 Hz). The results revealed that, in the zebrafish larvae exposed to 50 nm PS-NPs, the delta waves tended to decrease while the beta waves increased compared with the control group. However, in the zebrafish larvae exposed to 100 nm PS-NPs, there was no change in any EEG waves (Figure 2C). These data suggested that 50 nm PS-NPs, but not 100 nm PS-NPs, influence behavioral patterns and EEG waves in zebrafish larvae due to their accumulation in the nervous system.

### 2.3. Quantitative Profiling of Endogenous Neurochemicals in Polystyrene Nanoparticles (PS-NPs)-Exposed Zebrafish Larvae

We then profiled endogenous neurochemicals in zebrafish larvae using LC-MS/MS to investigate the relationship with behavioral and EEG patterns. Most of the neurochemicals were markedly changed in the zebrafish larvae expose to 50 nm PS-NPs compared with the control group. However, the 100 nm PS-NP group did not show any significant changes in the amount of neurochemicals (Figure 3A). In particular, dopaminergic metabolites including phenyl-alanine (Phe), tyrosine (Tyr), dopamine (DA), 3-methoxytyramine (3-MT), and octopamine (OA) decreased in the 50 nm PS-NPs group. The amount of 3-MT also decreased in the zebrafish larvae exposed to 100 nm PS-NPs, whereas other dopaminergic metabolite changes were not significant (Figure 3B). To determine whether the decreased dopaminergic activity was caused by the downregulation of enzymes related to dopaminergic metabolism, we analyzed the level of gene expression through quantitative real-time polymerase chain reaction (qRT-PCR). However, gene-related dopamine metabolism did not show any significant changes (Appendix A). Some GABAergic, cholinergic, and histaminergic metabolites also tended to decrease in the 50 nm PS-NP-exposed larvae. In particular, glutamine (Gln), glutamic acid (Glu), betaine (BET), and histamine (His) were significant (Appendix A). The two PS-NP groups did not affect the amount of serotonergic metabolites in the zebrafish larvae (Appendix A). These results suggest that accumulated PS-NPs in zebrafish larvae dysregulate endogenous neurochemicals, particularly dopaminergic metabolites.

### 2.4. Analysis of Polystyrene Nanoparticles (PS-NPs) Permeability on Blood–Brain Barrier (BBB) Chip

Our data represent that relatively small PS-NPs accumulate in the brain and cause disruption to the nervous system. However, if PS-NPs are absorbed into the skin or penetrate the BBB was not determined. The BBB is a unique microvasculature property in the brain that is a highly selective semipermeable border of endothelial cells to protect the brain from circulating exogenous and endogenous materials [25]. The BBB is composed of three cell types, endothelial cells, astrocytes, and pericytes, forming tight junctions as a barrier between endothelial cells [26]. To examine this issue, we mimicked an artificial BBB chip using human brain microvascular endothelial cells (HBMEC), human brain vascular pericytes (HBVP), and normal human astrocytes (NHA). The BBB chip consisted of two channels: (1) an HBMEC population mimicking blood vessels in the bottom channel; (2) HBVP and NHA populations mimicking the brain region in the top channel. We seeded cells into each channel, cultured them for 24 h, and then initiated a microfluidic system by introducing PS-NPs in the top channel (Figure 4A,B). To assess whether the PS-NPs penetrated the mimicked BBB, we measured fluorescent intensities in the bottom inlet and the top outlet. As a result of calculating the brain-to-blood ratio, it was found that the 50 nm PS-NPs were more able to permeate than the 100 nm PS-NPs (Figure 4C).

## 3. Discussion

Microplastics are one of the greatest threats to aquatic environments and can travel around the ecosystem. Their effect on human health is as yet unknown; however, accumulated microplastics in organs cause possible damage to human cells. In this study, we examined two issues related to predicting the dangers of nanoparticles on ecosystems using zebrafish larvae: (1) the differences between two sizes of nanoparticles accumulating in zebrafish larvae; and (2) the influence of accumulated nanoparticles on the central nervous system.

### 3.1. Exposure and Accumulation of Polystyrene Nanoparticles (PS-NPs) in Zebrafish Larvae

The accumulation of microplastics into the organs of zebrafish larvae tends to be size-dependent. Relatively smaller microplastics (0.02–0.2 μm) accumulate in internal organs such as the brain, eyes, liver, pancreas, and heart; however, larger microplastics (>0.2 μm) accumulate in the gut, gills, and skin [20]. We selected two sizes of fluorescent carboxylate polystyrene nanoparticles (PS-NPs), 50 and 100 nm, to visualize the accumulation of nanoparticles in transparent zebrafish larvae. As PS-NPs are absorbed through the skin only during the embryonic period, we exposed zebrafish larvae to PS-NPs during the larval stage when spontaneous food intake begins. Survival tests showed that 50 nm PS-NPs are more mortal than 100 nm (Figure 1B). Although our results showed that the sublethal dose is higher than in previous studies [20], it has been reported that highly accumulated, smaller-sized nanoparticles are more harmful [7]. For this reason, the effective concentration of PS-NPs was set at 1000 ppm in this study. Fluorescence of 50 nm PS-NPs was observed in the brain, intestine, and blood vessels, whereas 100 nm PS-NPs was not (Figure 1C,D). Next, we homogenized PS-NPs-exposed larvae to quantify the accumulated PS-NPs. As a result, 627.27 (±14.01) ppm was detected in the larvae exposed to 50 nm PS-NPs at 1000 ppm concentration, whereas 160.25 (±36.82) ppm was detected in the larvae exposed to 100 nm PS-NPs (Appendix A). Although we could not quantify PS-NPs in the brain parts, it was determined that the amount of PS-NPs in larvae was different depending on size. This result suggested that smaller-sized PS-NPs can accumulate larger amounts in the larvae.

### 3.2. Anxious Effects of Zebrafish Larvae Exposed to Polystyrene Nanoparticles (PS-NPs)

To determine how PS-NPs accumulated in the brain affect the nervous system, we analyzed behavioral patterns and electroencephalogram (EEG) in zebrafish larvae.

Locomotor activity following light and dark stimuli show a specific pattern depending on the transition—light to dark or dark to light [16]. The light-to-dark transition increased locomotor activity, which means an increased anxiety level regarding the new environment [27]. The 50 nm PS-NPs group tended to have an increased anxiety response during both stimuli, whereas it did not change in the 100 nm PS-NPs group (Figure 2A).

Color preference is an important trait that allows individuals to discriminate and prefer objects, such as foods, shoals, and predators. Color preference has been reported to affect the learning and memory behaviors of zebrafish [28]. Color preference tests using larval and adult zebrafish are well established [17,29], and there have been studies applying these tests to the evaluation of the efficacy and toxicity of drugs and pollutants [18,30]. In this study, the 50 nm PS-NPs group confused color preference compared with the control group (Figure 2B), which means that the PS-NPs had an influence on the central nervous system of zebrafish.

Analysis of larval EEGs has been used in the study of neurological disorders such as epilepsy to screen anti-seizure drugs. Single larva embedded in agarose was used to record EEG through a disease model; however, multiple larvae were used with an optimized microfluidic system in this study [19]. In general, brain waves are divided, depending on their frequency, into several sub-bands, being delta (1–4 Hz), theta (4–8 Hz), alpha (8–12 Hz), beta (12–30 Hz), and gamma (>30 Hz) [31]. This division based on power spectral analysis is a well-established method used in the analysis of EEG signals [32]. In the larvae exposed to 50 nm PS-NPs, the delta waves tended to decrease, and these are the waves that prominently occur in deep sleep in normal people or newborns. Beta waves, which usually appear when we are awake, speaking, feeling tension, or experiencing anxiety, also showed an increase in the 50 nm PS-NPs-exposed zebrafish larvae (Figure 2C).

The EEG data can be interpreted as showing that the increase in beta waves and the decrease in delta waves were consistent with anxious behaviors in the 50 nm PS-NPs-exposed zebrafish larvae.

### 3.3. Dopaminergic Dysregulation of Zebrafish Larvae Exposed to Polystyrene Nanoparticles (PS-NPs)

Quantifying endogenous neurochemicals from brain tissue using LC-MS/MS analysis provides much information for animal behavioral analysis. Neurotransmitters are chemical messengers associated with all of brain activities, and dopaminergic, serotonergic, cholinergic, and GABAergic neurotransmitters are well known [18].

Dopaminergic pathways are involved in motivated behaviors, various types of reward, and cognition processes. If the pathway is dysregulated in humans, neurodegenerative diseases such as Parkinson’s disease are caused [33,34]. Alterations in the dopaminergic system have also been implicated in anxiety and compulsive disorders [35]. In this study, we profiled 31 neurochemicals, among which dopaminergic metabolites, including Phe, Tyr, L-DOPA, DA, 3-MT, and OA, were significantly decreased in the 50 nm PS-NPs-exposed zebrafish larvae, suggesting that dopaminergic system activity was reduced (Figure 3). In general, it was well known that altered dopaminergic activity induces behavioral changes by use of agonists and antagonists. Dopamine receptor agonists such as cabergoline, bromocriptine, and quinpirole stimulate locomotor activity in MPTP-treated mice [36], whereas antagonists decrease locomotor activity in rat [37,38]. However, it has been reported that dopamine regulation using agonists and antagonists in animal models vary in behavioral profiles depending on concentration [36,37,38]. In addition, Irons et al. demonstrated that dopaminergic drugs induce differential locomotor activities following light and dark transition in zebrafish larvae [39]. For example, selective agonists induced hyperactivity, whereas a non-selective agonist, apomorphine, showed biphasic changes in locomotor activity [39]. Similarly, selective antagonists induced hypoactivity, whereas non-selective antagonist, butaclamol, showed biphasic locomotor activity [39]. Although it is inconsistent with decreased spontaneous locomotion in dopamine deficiency using antagonists, we demonstrated that endogenous dopaminergic antagonism induces hyperactivity following stimuli such as light and dark. This result is consistent with previous data that non-selective antagonism biphasically increases locomotor activity following light and dark stimuli [39].

In this study, we revealed that PS-NPs induce dopaminergic dysfunction; however, we could not suggest its mechanism. We attempted analysis of gene expression for dopaminergic metabolite-related enzymes using qRT-PCR to resolve this issue. However, PS-NPs did not affect the expression of dopamine-related enzymes (Appendix A). From this result, we expect that accumulated PS-NPs has the potential to inhibit enzymatic activity. In particular, we expect inhibition of DOPA decarboxylase, an enzyme that converts from L-DOPA to dopamine. We also expect that reduced dopamine affects the level of its metabolite, 3-MT. Although this study did not reveal how PS-NPs affect the dopamine system, we will perform further studies on enzymatic activity and neurotoxicity.

Taken together, we concluded that anxious behavior, recognition problems, and EEG alterations in the 50 nm PS-NPs-exposed zebrafish larvae were caused by decreased dopaminergic activity.

### 3.4. Blood-Brain Barrier (BBB) Permeability of Polystyrene Nanoparticles (PS-NPs)

The blood–brain barrier (BBB) plays a crucial role in maintaining the internal surroundings inside the brain and providing protection against invasion by harmful agents [25]. Brain vessel development in zebrafish is initiated at the embryonic stage, and cellular maturation organizing the BBB is progressed during larvae development [40]. Several studies have been performed to analyze the function of the BBB in larvae, with fluorescent tracers injected into the heart or caudal vein and assessed in the brain [41,42]. Adult zebrafish are also used to assess drug delivery into the brain using LC-MS/MS analysis [43]. We hypothesized that accumulated PS-NPs in the brain penetrate through the BBB. We then microinjected both PS-NPs into the caudal vein; however, no fluorescent signal could be found in the zebrafish larvae. It is presumed that the PS-NPs microinjected into the zebrafish larvae were probably too small in quantity to be detected using a confocal microscope. Since the BBB permeability of PS-NPs could not be evaluated using zebrafish larvae, we applied an organ-on-a-chip system to mimic the BBB. This BBB chip was a microfluidic system that consisted of two channels: the mimicked blood vessel (bottom) and brain (top). Although the two sizes of PS-NPs were detected in the top channel through penetration of the mimicked BBB, the 50 nm PS-NPs showed a relatively greater ability to permeate than the 100 nm PS-NPs. Some previous studies have assumed that nanoparticles may penetrate the BBB, and the penetrative size is expected to be under 2 μm [44,45]. Based on the mimicked BBB result, we concluded that nanoparticles can size-dependently penetrate the BBB.

## 4. Materials and Methods

### 4.1. Maintenance of Zebrafish

Adult zebrafish were maintained under standard conditions as previously described [46]. Fertilized eggs were collected after natural spawning and screened using a stereomicroscope to exclude unfertilized eggs. Embryonic media (60 μg/L of sea salt; Sigma-Aldrich, St. Louis, MO, USA) were replaced daily until 5 days post-fertilization (dpf). Experiments involving zebrafish were performed in accordance with the NIH Guide for the Care and Use of Laboratory Animals (no. 8023, revised in 1996), and this work was approved by the Animal Care and Use Committee of the Korea Research Institute of Chemical Technology (7B-ZF1).

### 4.2. Treatment of Polystyrene Nanoparticles (PS-NPs)

Fluorescent monodispersed PS-NPs were commercially purchased from Polysciences, Inc. (Warrington, PA, USA): Fluoresbrite^®^ YG Carboxylate Microspheres 0.05 μm (16661-10); Fluroesbrite^®^ YG Carboxylate Microspheres 0.1 μm (16662-10). Wild type were used for analysis of survival rate, behaviors, quantitative RT-PCR, EEG, and endogenous neurotransmitters. The blood vessel specific transgenic zebrafish line *Tg(kdrl:mCherry)* with *nacre* background were used for larval image. Furthermore, 5 dpf of wild-type or *Tg(kdrl:mCherry)* larvae were placed in 6-well plate (20 larvae per well) and exposed for a period of 96 h to PS-NPs, prepared by dilution of the stock suspension in embryonic media. Larvae incubated with only embryonic media served as the control group. All PS-NPs-exposed larvae were washed three times in embryonic media before each experiment. Survival rate of larvae was counted every day. Accumulation of PS-NPs in larvae was observed under K1-Fluo confocal fluorescence laser scanning microscopy (Nanoscope Systems, Daejeon, Korea).

### 4.3. Quantification of Polystyrene Nanoparticles (PS-NPs) in Zebrafish Larvae

Standard was generated by 0, 200, 400, 600, 800, and 1000 ppm PS-NPs exposed to zebrafish larvae for 72 h. Larvae were homogenized by CelLytic M buffer (C2978, Sigma-Aldrich) after exposing PS-NPs. Then, 200 μL of homogenates was added into each well in BD Falcon 96-well Flat Bottom Microplate (353376, Corning, NY, USA). PS-NPs concentration was calculated as a fluorescence intensity (Ex/Em = 441/510 nm) that was measured using a microplate reader (M100PRO, TECAN, Zürich, Switzerland).

### 4.4. Behavioral Analysis of Zebrafish Larvae

Larvae were placed in 96-well plate with 200 μL of embryonic media. Behavioral analysis following light/dark stimuli was used by DanioVision Observation Chamber with EthoVision XT 15 (Noldus, The Netherlands). Larvae in 96-well plate were acclimated to the dark for 20 min and stimulated to six cycles of 10 min of light and dark, respectively. Locomotor patterns following stimuli were represented as distance moved (mm) per 1 min. Locomotor activities were represented as percentage of control group during each stimuli for 10 min (*n* = 12, triplicates).

Larvae were placed in a 2-channel (blue and yellow) chamber with 10 mL embryonic media. Twenty larvae per group were recorded for 30 min by a digital camera (HDR-CX130, Sony, Japan) mounted 50 cm above the apparatus. Images were acquired from recorded movie per 2 min. Then, each color preference was calculated as a mean of channel in which larvae were located in the acquired images.

### 4.5. Analysis of Electroencephalogram (EEG)

EEG was performed by Zefit (Daegu, Korea) to determine brain activity following PS-NPs in zebrafish larvae. Analysis of EEG using microfluidic system was carried out according to a previous study [19].

### 4.6. Analysis of Endogenous Neurochemicals

The quantification and analysis of endogenous neurochemicals was performed according to a previously described study [18]. Briefly, PS-NPs-exposed larvae (30 pooled, six experimental times) were added to distilled water and homogenized by ultra-sonication. Each lysate was centrifuged at 15,000 rpm for 20 min at 4 °C, and supernatants were analyzed by an Ultra-Performance Liquid Chromatography (ACQUITY UPLC System, Waters Corporation, Milford, MA, USA) coupled with a Xevo TQ-S Triple Quadrupole Mass Spectrometer (Waters Corporation).

### 4.7. Quantitative Real-Time Polymerase Chain Reaction (qRT-PCR)

Total RNA was isolated from zebrafish larvae (20 pooled) after exposure of PS-NPs using TRIzol reagent (15596026, Invitrogen, Waltham, MA, USA) and purified according to manufacturer’s protocol. qRT-PCR was performed using Verso SYBR Green 1-Step qRT-PCR Low ROX Mix (AB-4106/A, Thermo Scientific, Waltham, MA, USA). Gene expression level was analyzed for relative change using the 2^−ΔΔCT^ method and normalized to β-actin as an endogenous control. Primer sets are listed in Appendix A.

### 4.8. Cell Culture

Immortalized human brain microvascular endothelial cells (HBMEC; #1000, ScienCell, Carlsbad, CA, USA) were maintained in endothelial cell medium (#1001, ScienCell). Human brain vascular pericytes (HBVP; #1200, ScienCell) and normal human astrocytes (NHA; CC-2565, Lonza, Basel, Switzerland) were maintained in astrocyte (CC-3186, Lonza) and pericyte medium (#1201, ScienCell), respectively. All cells were cultured at 37 °C in a humidified chamber containing 95% air and 5% CO_2_. Before seeding cells into the BBB chip, HBMEC, HBVP, and NHA were stained with CellTracker™ Green CMFDA dye (C2925, Invitrogen), Orange CMTMR dye (C2927, Invitrogen), and Blue CMAC dye (C2110, Invitrogen), respectively. Stained cells were imaged by Lionheart FX Automated Microscope (BioTek Instruments, Winooski, VT, USA).

### 4.9. Construction of Blood-Brain Barrier (BBB) Chip System

The design and construction of BBB chip was based on previously reported protocols [47,48]. The chip (OCK-12, Emulate, Inc., Boston, MA, USA) was composed of two parallel micro-channels (top: 1 × 1 mm^2^; bottom: 1 × 0.2 mm^2^), which were separated by a porous polydimethylsiloxane (PDMS) membrane (diameter: 7 nm; spacing: 40 nm; thickness: 50 nm). PDMS membranes were activated by manufacturer’s instruction and coated with a mixture of collagen I (100 mg/mL) and fibronectin (0.025 mg/mL) diluted with Dulbecco’s phosphate-buffered saline (DPBS). Coated chips were incubated at 4 °C overnight, then at 37 °C for 1 h. HBMECs were detached from the cell culture plate and seeded into the bottom channel of the chip with a density of 9 × 105 cells/mL. The chip was immediately inverted and incubated at 37 °C for 1 h. After 4 h, HBMECs were prepared by previous method and injected into the bottom channel once again. HBVP and NHA were detached from the cell culture plate, and the cells were mixed to a density of 0.04 × 105 and 0.4 × 105 cells/mL, respectively. The cell mixture was introduced into the top channel. The chip was incubated at 37 °C overnight and then attached to a Pod^®^ Portable Module (Emulate). Endothelial cell medium (3 mL) containing 100 ppm PS-NPs was filled in an inlet reservoir for bottom channel. The mixture (3 mL) of astrocyte and pericyte medium (1:1) was filled in an inlet reservoir for the top channel. Pods were loaded on the Zoë^®^ (Emulate) culture module and underwent a regulate cycle followed by a constant flow rate of 30 μL/hour in bottom channel and 80 μL/hour in top channel. Media collected in respective inlet and outlet reservoir were used for fluorescence measurements. The fluorescence intensity (Ex/Em = 441/510 nm) was measured using a microplate reader (M100PRO, TECAN).

### 4.10. Statistical Analysis

All statistical data were presented as means ± standard error of mean (SEM) using Excel 2016 (Microsoft Corporation, Redmond, WA, USA) and GraphPad Prism 9.4.0 (GraphPad Software, Inc., San Diego, CA, USA). Analysis of data sets was performed using the Mann–Whitney *U* test, and statistical significance was set at 0.05, 0.01, and 0.001 (* *p* ≤ 0.05, ** *p* ≤ 0.01, and *** *p* ≤ 0.001).

## 5. Conclusions

In this study, we showed that polystyrene nanoparticles (PS-NPs) can penetrate a mimicked blood–brain barrier (BBB) using a microfluidic system, and that PS-NPs accumulate in the brains of zebrafish larvae. Moreover, electroencephalogram (EEG) analysis showed that PS-NPs caused abnormal behavioral patterns due to dopaminergic metabolism dysregulation in zebrafish larvae. Taken together, we suggest that an excessive accumulation of nanoparticles in the central nervous system can induce neurological disorders such as dopaminergic dysregulation.

## Figures and Tables

**Figure 1 ijms-23-10682-f001:**
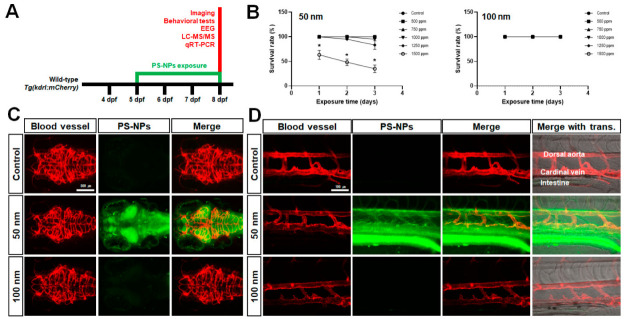
Accumulation of polystyrene nanoparticles (PS-NPs) in zebrafish larvae. (**A**) Experimental scheme of PS-NPs-exposed zebrafish larvae. (**B**) Survival test in PS-NPs-exposed zebrafish larvae. Representative images of the brain (**C**) and the posterior trunk region (**D**) in 1000 ppm of PS-NPs-exposed *Tg(kdrl:mCherry)*; *mitfa^−/−^*. Red and green fluorescence indicate the blood vessel and accumulated PS-NPs in the brain (**C**) and the posterior trunk (**D**), respectively. Data sets were expressed as the means ± standard error of the mean and statistical significance was set at 0.05, 0.01, and 0.001 (* *p* ≤ 0.05).

**Figure 2 ijms-23-10682-f002:**
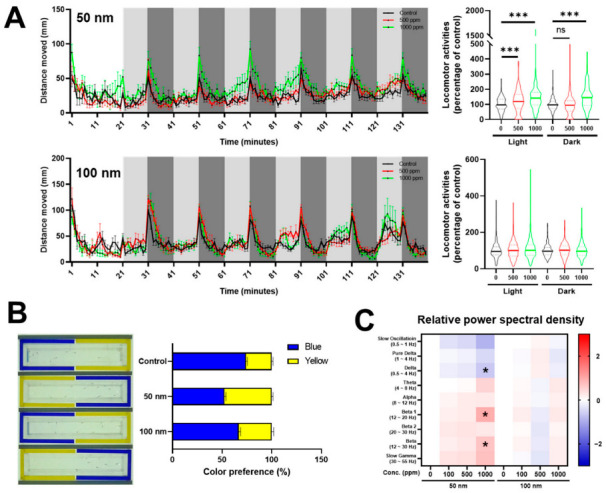
Behavioral and electroencephalogram (EEG) analysis of polystyrene nanoparticles (PS-NPs)-exposed zebrafish larvae. (**A**) Light and dark transition test. Experimental concentration of both PS-NPs were 1000 ppm. (**B**) Color preference test in blue–yellow chamber. (**C**) Heatmap of relative power spectral density through EEG analysis. The normalized Z-scores were calculated by the mean of the population and standard deviation of the population. Data sets were expressed as the means ± standard error of the mean and statistical significance was set at 0.05, 0.01, and 0.001 (* *p* ≤ 0.05, *** *p* ≤ 0.001, ns—not statistically significant).

**Figure 3 ijms-23-10682-f003:**
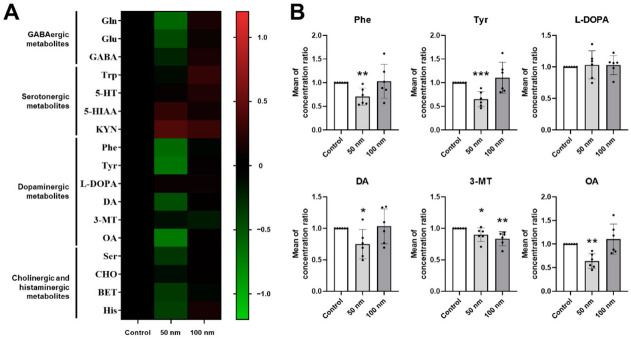
Quantitative analysis of endogenous neurochemicals in polystyrene nanoparticles (PS-NPs)-exposed zebrafish larvae. (**A**) Heatmap of endogenous neurochemicals relative to the control. The normalized Z-scores were calculated by the mean of the concentration ratio and standard deviation in each experimental group. Experimental concentration of both PS-NPs was 1000 ppm. (**B**) The changes of dopaminergic metabolites in PS-NPs-exposed zebrafish larvae. Each metabolite was calculated as the mean of concentration ratio that was expressed as the ratio of the neurotransmitter compared to the control in each experimental groups. Data sets were expressed as the means ± standard error of the mean and statistical significance was set at 0.05, 0.01, and 0.001 (* *p* ≤ 0.05, ** *p* ≤ 0.01, and *** *p* ≤ 0.001). 3-MT: 3-methoxytyramine; 5-HIAA: 5-hydroxyindoleacetic acid; 5-HT: 5-hydroxytryptamine (serotonin); BET: betaine; CHO: choline; DA: dopamine; GABA: gamma-aminobutyric acid; Gln: glutamine; Glu: glutamic acid; His: histidine; KYN: kynurenine; L-DOPA: 3,4-dihydroxy-L-phenylalanine; OA: octopamine; Phe: phenylalanine; Ser: serine; Trp: tryptophan; Tyr, Tyrosine.

**Figure 4 ijms-23-10682-f004:**
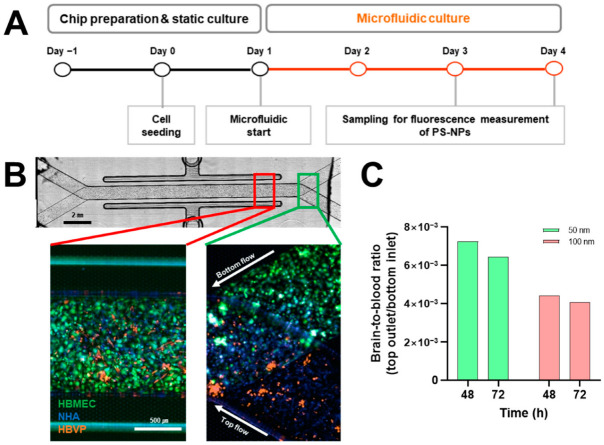
Analysis of polystyrene nanoparticles (PS-NPs) permeability on blood–brain barrier (BBB) chip. (**A**) Experimental scheme of permeability test on BBB chip. Experimental concentration of both PS-NPs was 100 ppm. (**B**) Images of BBB chip and stained cells. (**C**) Test for permeability of two PS-NPs through BBB chip. Brain-to-blood ratio was calculated as relative fluorescence unit (RFU) of outlet chamber in top channel (brain)-to-inlet chamber in bottom channel (blood) ratio. HBMEC: human brain microvascular endothelial cells; HBVP: human brain vascular pericytes; NHA: normal human astrocytes.

## Data Availability

Not applicable.

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
