# Peer review of "Size-Dependent Effects of Polystyrene Nanoparticles (PS-NPs) on Behaviors and Endogenous Neurochemicals in Zebrafish Larvae"

_ijms, 2022, doi:10.3390/ijms231810682_

Round 1
Reviewer 1 Report
This manuscript addresses the effects of plastic nanoparticles in embryonic zebrafish. It is very complete. However, there are several points in the discussion that need to be addressed: (1) The discussion would benefit from subsections; (2) the authors equate increased locomotion with an anxiolytic response; Without more detailed analyses as to where in the well the embryos move, increased locomotion could be convulsive movements (Torres-Hernandez, et al, 2016, Do Carmo Silva et al , 2018); Decreased DA levels, without changes in L-DOPA implicate delayed or inhibited DA transmission. Similarly 3MT needs to be discussed further, even if other models are used. The authors need to consider that their generous reference to Parkinson, needs to be thoroughly documented given that these are embryonic zebrafish
Reviewer 2 Report
I have received the research article "Size-dependent effects of polystyrene nanoparticles (PS-NPs) on behaviors and endogenous neurochemicals in zebrafish larvae" by Kyu-Seok Hwang et al. for evaluation.
In their study, the authors used the zebrafish model to explore the neurotoxic effect of polystyrene nanoparticles on the locomotor activity and neurotransmitter system in the brain. The study concludes that small nanoparticles (50 nm) have high BBB penetration ability which is responsible for compromised dopaminergic system and abnormal locomotor activity.
The work is timely and important in building on growing evidence to decipher the neurotoxic effect of polystyrene nanoparticles and create awareness against this environmental pollutant. The study is novel and experiments are well planned and executed nicely. However, I have some critical comments about this study. My specific comments are mentioned below:-
Major comments:
1. Authors have shown that exposure to small nanoparticles (50 nm) suppresses the activity of the dopaminergic system in the brain of zebrafish. However, the study failed to provide the root cause of dopaminergic system impairment.
2. In the behavioral experiment, authors have shown that exposure to small nanoparticles (50 nm) increases locomotors' activity. The behavioral result contradicts the finding showing the under activity of the dopaminergic system in the brain of small nanoparticles (50 nm) exposed zebrafish.
3. Author has adapted the qualitative method to show the accumulation of small nanoparticles (50 nm) shown in the brain. I would suggest quantifying the concentration of small nanoparticles (50 nm) in the brain.
Minor comments:
1. Parametric statistical tests (Student’s T-test) are not acceptable for small samples. I would recommend adopting nonparametric statistical tests such as the Mann-Whitney test (unpaired comparisons).
2. In the Statistical analysis, the author mentioned in line number 382 that statistical significance was set at 50, 0.01, and 0.001 (*p ≤ 50, **p ≤ 0.01, and ***p ≤ 0.001). What is the meaning of 50 and *p ≤ 50?
Round 2
Reviewer 2 Report
Accepted